# LIKELIHOOD CONTRIBUTION BASED MULTI-SCALE ARCHITECTURE FOR GENERATIVE FLOWS

## ABSTRACT

Deep generative modeling using flows has gained popularity owing to the tractable exact log-likelihood estimation with efficient training and synthesis process. However, flow models suffer from the challenge of having high dimensional latent space, same in dimension as the input space. An effective solution to the above challenge as proposed by Dinh et al. (2016) is a multi-scale architecture, which is based on iterative early factorization of a part of the total dimensions at regular intervals. Prior works on generative flows involving a multi-scale architecture perform the dimension factorization based on a static masking. We propose a novel multi-scale architecture that performs data dependent factorization to decide which dimensions should pass through more flow layers. To facilitate the same, we introduce a heuristic based on the contribution of each dimension to the total log-likelihood which encodes the importance of the dimensions. Our proposed heuristic is readily obtained as part of the flow training process, enabling versatile implementation of our likelihood contribution based multi-scale architecture for generic flow models. We present such implementations for several state-of-the-art flow models and demonstrate improvements in log-likelihood score and sampling quality on standard image benchmarks. We also conduct ablation studies to compare proposed method with other options for dimension factorization.

## 1 INTRODUCTION

Deep Generative Modeling aims to learn the embedded distributions and representations in input (especially unlabelled) data, requiring no/minimal human labelling effort. Learning without knowledge of labels (unsupervised learning) is of increasing importance because of the abundance of unlabelled data and the rich inherent patterns they posses. The representations learnt can then be utilized in a number of downstream tasks such as semi-supervised learning (Kingma et al., 2014; Odena, 2016), synthetic data augmentation and adversarial training (Cisse et al., 2017), text analysis and model based control etc. The repository of deep generative modeling majorly includes Likelihood based models such as autoregressive models (Oord et al., 2016b; Graves, 2013), latent variable models (Kingma & Welling, 2013), flow based models (Dinh et al., 2014; 2016; Kingma & Dhariwal, 2018) and implicit models such as generative adversarial networks (GANs) (Goodfellow et al., 2014). Autoregressive models (Salimans et al., 2017; Oord et al., 2016b;a; Chen et al., 2017) achieve exceptional log-likelihood score on many standard datasets, indicative of their power to model the inherent distribution. But, they suffer from slow sampling process, making them unacceptable to adopt in real world applications. Latent variable models such as variational autoencoders (Kingma & Welling, 2013) tend to better capture the global feature representation in data, but do not offer an exact density estimate. Implicit generative models such as GANs which optimize a generator and a discriminator in a min-max fashion have recently become popular for their ability to synthesize realistic data (Karras et al., 2018; Engel et al., 2019). But, GANs do not offer a latent space suitable for further downstream tasks, nor do they perform density estimation.

Flow based generative models (Dinh et al., 2016; Kingma & Dhariwal, 2018) perform exact density estimation with fast inference and sampling, due to their parallelizability. They also provide an information rich latent space suitable for many applications. However, the dimension of latent space for flow based generative models is same as the high-dimensional input space, by virtue of bijectivity nature of flows. This poses a bottleneck for flow models to scale with increasing input dimensions due to computational complexity. An effective solution to the above challenge is a multi-scale architecture,

introduced by Dinh et al. (2016), which performs iterative early gaussianization of a part of the total dimensions at regular intervals of flow layers. This not only makes the model computational and memory efficient but also aids in distributing the loss function throughout the network for better training. Many prior works including Kingma & Dhariwal (2018); Atanov et al. (2019); Durkan et al. (2019); Kumar et al. (2019) implement multi-scale architecture in their flow models, but use static masking methods for factorization of dimensions. We propose a multi-scale architecture which performs data dependent factorization to decide which dimensions should pass through more flow layers. For the decision making, we introduce a heuristic based on the amount of total log-likelihood contributed by each dimension, which in turn signifies their individual importance. We lay the ground rules for quantitative estimation and qualitative sampling to be satisfied by an ideal factorization method for a multi-scale architecture. Since in the proposed architecture, the heuristic is obtained as part of the flow training process, it can be universally applied to generic flow models. We present such implementations for flow models based on affine/additive coupling and ordinary differential equation (ODE) and achieve quantitative and qualitative improvements. We also perform ablation studies to confirm the novelty of our method. Summing up, the contributions of our research are,

**Contributions:**

1. A log-determinant based heuristic which entails the contribution by each dimensions towards the total log-likelihood in a multi-scale architecture.
2. A multi-scale architecture based on the above heuristic performing data-dependent splitting of dimensions, implemented for several classes of flow models.
3. Quantitative and qualitative analysis of above implementations and an ablation study

To the best of our knowledge, we are the first to propose a data-dependent splitting of dimensions in a multi-scale architecture.

## 2 BACKGROUND

In this section, we illustrate the functioning of flow based generative models and the multiscale architecture as introduced by Dinh et al. (2016).

### 2.1 FLOW-BASED GENERATIVE MODELS

Let $\mathbf{x}$ be a high-dimensional random vector with unknown true distribution $p(\mathbf{x})$. The following formulation is directly applicable to continous data, and with some pre-processing steps such as dequantization (Uria et al., 2013; Salimans et al., 2017; Ho et al., 2019) to discrete data. Let $\mathbf{z}$ be the latent variable with a known standard distribution $p(\mathbf{z})$, such as a standard multivariate gaussian. Using an i.i.d. dataset $\mathcal{D}$, the target is to model $p_{\boldsymbol{\theta}}(\mathbf{x})$ with parameters $\boldsymbol{\theta}$. A flow, $\mathbf{f}_{\boldsymbol{\theta}}$ is defined to be an invertible transformation that maps observed data $\mathbf{x}$ to the latent variable $\mathbf{z}$. A flow is invertible, so the inverse function $\mathcal{T}$ maps $\mathbf{z}$ to $\mathbf{x}$, i.e.

$$\mathbf{z} = \mathbf{f}_{\boldsymbol{\theta}}(\mathbf{x}) = \mathcal{T}^{-1}(\mathbf{x}) \ \text{ and } \ \mathbf{x} = \mathcal{T}(\mathbf{z}) = \mathbf{f}_{\boldsymbol{\theta}}^{-1}(\mathbf{z}) \tag{1}$$

The log-likelihood can be expressed as,

$$\log\left(p_{\boldsymbol{\theta}}(\mathbf{x})\right) = \log(p(\mathbf{z})) + \log\left(\left|\det\left(\frac{\partial \mathbf{f}_{\boldsymbol{\theta}}(\mathbf{x})}{\partial \mathbf{x}^T}\right)\right|\right) \tag{2}$$

where $\dfrac{\partial \mathbf{f}_{\boldsymbol{\theta}}(\mathbf{x})}{\partial \mathbf{x}^T}$ is the Jacobian of $\mathbf{f}_{\boldsymbol{\theta}}$ at $\mathbf{x}$.

The invertibile nature of flow allows it to be capable of being composed of other flows of compatible dimensions. In practice, flows are constructed by composing a series of component flows. Let the flow $\mathbf{f}_{\boldsymbol{\theta}}$ be composed of $K$ component flows, i.e. $\mathbf{f}_{\boldsymbol{\theta}} = \mathbf{f}_{\theta_K} \circ \mathbf{f}_{\theta_{K-1}} \circ \cdots \circ \mathbf{f}_{\theta_1}$ and the intermediate variables be denoted by $\mathbf{y}_K, \mathbf{y}_{K-1}, \cdots, \mathbf{y}_0 = \mathbf{x}$. Then the log-likelihood of the composed flow is,

$$\log\left(p_{\boldsymbol{\theta}}(\mathbf{x})\right) = \log(p(\mathbf{z})) + \log\left(\left|\det\left(\frac{\partial(\mathbf{f}_{\theta_K} \circ \mathbf{f}_{\theta_{K-1}} \circ \cdots \circ \mathbf{f}_{\theta_1}(\mathbf{x}))}{\partial \mathbf{x}^T}\right)\right|\right) \tag{3}$$

$$= \underbrace{\log(p(\mathbf{z}))}_{\text{Log-latent density}} + \sum_{i=1}^{K} \underbrace{\log|\det(\partial \mathbf{y}_i / \partial \mathbf{y}_{i-1}^T)|}_{\text{Log-det}} \tag{4}$$

which follows from the fact that $\det(A \cdot B) = \det(A) \cdot \det(B)$. In our work, we refer the first term in Equation 4 as *log-latent-density* and the second term as *log-determinant (log-det)*. The reverse path, from $\mathbf{z}$ to $\mathbf{x}$ can be written as a composition of inverse flows, $\mathbf{x} = \mathbf{f}_{\boldsymbol{\theta}}^{-1}(\mathbf{z}) = \mathbf{f}_{\theta_1}^{-1} \circ \mathbf{f}_{\theta_2}^{-1} \circ \cdots \circ \mathbf{f}_{\theta_K}^{-1}(\mathbf{z})$. Confirming with the properties of a flow as mentioned above, different types of flows can be constructed (Kingma & Dhariwal, 2018; Dinh et al., 2016; 2014; Behrmann et al., 2018).

## 2.2 MULTI-SCALE ARCHITECTURE

Multi-scale architecture is a design choice for latent space dimensionality reduction of flow models, in which part of the dimensions are factored out/early gaussianized at regular intervals, and the other part is exposed to more flow layers. The process is called dimension factorization. In the problem setting as introduced in Section 2.1, the factoring operation can be mathematically expressed as,

$$\mathbf{y}_0 = \mathbf{x} \tag{5}$$
$$\mathbf{z}_{l+1}, \mathbf{y}_{l+1} = \mathbf{f}_{\theta_{l+1}}(\mathbf{y}_l), \quad l \in \{0, 1, \cdots, K-2\} \tag{6}$$
$$\mathbf{z}_K = \mathbf{f}_{\theta_K}(\mathbf{y}_{K-1}) \tag{7}$$
$$\mathbf{z} = (\mathbf{z}_1, \mathbf{z}_2, \cdots, \mathbf{z}_K) \tag{8}$$

The factoring of dimensions at early layers has the benefit of distributing the loss function throughout the network (Dinh et al., 2016) and optimizing the amount of computation and memory used by the model. We consider the multi-scale architecture for flow based generative models as introduced by Dinh et al. (2016) (and later used by state-of-the-art flow models such as Glow(Kingma & Dhariwal, 2018)) as the base of our research work.

## 3 LIKELIHOOD CONTRIBUTION BASED MULTISCALE ARCHITECTURE

In a multi-scale architecture, it is apparent that the network will better learn the distribution of variables getting exposed to more layers of flow as compared to the ones which get factored at a finer scale (earlier layer). The method of dimension splitting proposed by prior works such as (Dinh et al., 2016; Kingma & Dhariwal, 2018; Behrmann et al., 2018) are static in nature and do not distinguish between importance of different dimensions. In this section, we introduce a heuristic to estimate the contribution of each dimension towards the total log-likelihood, and introduce a method which can use the heuristic to decide the dimensions to be factored at an earlier layer, eventually achieving preferrential splitting in multiscale architecture. Our approach builds an efficient multiscale architecture which factors the dimensions at each flow layer in a way such that the local variance in the input space is well captured as the flow progresses and the log-likelihood is maximized. We also describe how our multi-scale architecture can be implemented over several standard flow models.

Recall from Equation 4 that the log-likelihood is composed of two terms, the log-latent-density term and the log-det term. The log-latent-density term depends on the choice of latent distribution whereas the log-det term depends on the modeling of the flow layers. So, careful design of flow layers can lead to maximized log-determinant, eventually maximizing the likelihood. The total log-det term is nothing but the sum of log-det terms contributed by each dimension. Let the dimension of the input space $\mathbf{x}$ be $s \times s \times c$, where $s$ is the image height/width and $c$ is the number of channels for image inputs. For the following formulation, let us assume no dimensions were gaussianized early so that we have access to log-det term for all dimensions at each flow layer, and the dimension at all intermediate layer remains same (i.e. $s \times s \times c$). We apply a flow ($\mathbf{f}_{\boldsymbol{\theta}}$) with $K$ component flows to $\mathbf{x}$, $\mathbf{z}$ pair, so that $\mathbf{z} = \mathbf{f}_{\boldsymbol{\theta}}(\mathbf{x}) = \mathbf{f}_{\theta_K} \circ \mathbf{f}_{\theta_{K-1}} \circ \cdots \circ \mathbf{f}_{\theta_1}(\mathbf{x})$. The intermediate variables are denoted by $\mathbf{y}_K, \mathbf{y}_{K-1}, \cdots, \mathbf{y}_0$ with $\mathbf{y}_K = \mathbf{z}$ (since no early gaussianization was done) and $\mathbf{y}_0 = \mathbf{x}$. The log-det term at layer $l$, $L_d^{(l)}$, is given by,

$$[L_d^{(l)}]_{scaler} = \sum_{i=1}^{l} \log |\det(\partial \mathbf{y}_i / \partial \mathbf{y}_{i-1}^T)| \tag{9}$$

The log-det of the jacobian term encompasses contribution by all the $s \times s \times c$ dimensions. We decompose it to obtain the individual contribution by variables (dimensions) towards the total log-det ($\sim$ total log-likelihood). The log-det term can be viewed (with slight abuse of notations) as a $s \times s \times c$

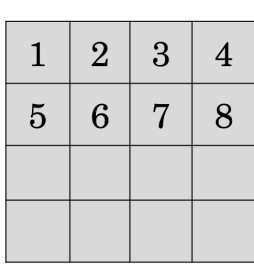 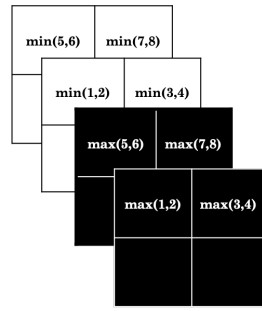

Figure 1: Likelihood contribution based squeezing operation for RealNVP: (On left) The tensor $[L_d^{(l)}]_{s \times s \times c}$ representing log-det of variables in a flow layer. (On right) It is squeezed to $\frac{s}{2} \times \frac{s}{2} \times 4c$ with local max and min pooling operation. The black (respectively white) marked pixels represent dimensions having more (respectively less) log-det locally.

tensor corresponding to each of the dimensions, summed over the flow layers till $l$,

$$[L_d^{(l)}]_{s \times s \times c} = \sum_{i=1}^{l} [d_{i-1}^{(\alpha, \beta, \gamma)}]_{s \times s \times c}, \quad \text{where} \quad \alpha, \beta \in \{0, \cdots, s\} \quad \text{and} \quad \gamma \in \{0, \cdots, c\} \tag{10}$$

$$\text{s.t.} \quad \sum_{\alpha, \beta, \gamma} d_{i-1}^{(\alpha, \beta, \gamma)} = \log|\det(\partial \mathbf{y}_i / \partial \mathbf{y}_{i-1}^T)| \tag{11}$$

The entries in $[L_d^{(l)}]_{s \times s \times c}$ having higher value correspond to the variables which contribute more towards the total log-likelihood, hence are more valuable for better flow formulation. So, we can use the *likelihood contribution (in the form of log-det term) by each dimension* as a heuristic for deciding which variables should be gaussianized early in a multi-scale architecture. Ideally, at each flow layer, the variables with more log-det term should be exposed to more layer of flow and the ones having less log-det term should be factored out. In this manner, selectively more power can be provided to variables which capture meaningful representation (and are more valuable from log-det perspective) to be expressive by being exposed to multiple flow layers. This formulation leads to enhanced density estimation performance. Additionally, for many datasets such as images, the spatial nature should be taken into account while deciding dimensions for early gaussianization. Summarily, at every flow layer, an ideal factorization method should,

1. *(Quantitative) For efficient density estimation:* Early gaussianize the variables having less log-det and expose the ones having more log-det to more flow layers

2. *(Qualitative) For qualitative reconstruction:* Capture the local variance over the flow layers, i.e. the dimensions being exposed to more flow layers should contain representative pixel variables from throughout the whole image.

Keeping the above requirements in mind, variants of hybrid techniques for factorization can be implemented for different types of flow models which involve a multi-scale architecture, to improve their density estimation and qualitative performance. The key requirement is availability of log-det contributions per dimension, which can be fulfilled by decomposition of the log-det of the jacobian. We refer to the method as Likelihood Contribution based Multi-scale Architecture (LCMA). The steps of LCMA implementation for flow models is summarized in Algorithm 1. Note that in step 2 of dimension factorization phase in algorithm 1, we group the dimensions having more/less log-det locally and then perform splitting. This preserves the local spatial variation of the image in both parts of the factorization, leveraging both enhanced density estimation as well as qualitative reconstruction. Another important observation is since the factorization of dimensions does not occur during the training time, and before the actual training starts, the decision of dimensions which get factored at each flow layer is fixed, the change of variables formula can be applied. This allows the use of non-invertible operations (e.g. max and min pooling) for efficient factorization with log-det heuristic.

Step 1 of dimension factorization phase requires computation of individual contribution of dimensions ($[L_d^{(l)}]_{s \times s \times c}$) towards the total log-likelihood, which can vary depending on the original design of flow

---

**Algorithm 1:** LCMA implementation for generative flow models

---

**Pre-Training Phase:** Pre-train a network with no multiscale architecture (no dimensionality reduction) to obtain the log-det term at every flow layer.

**Dimension Factorization:** In this phase, the dimensions to be factored at each flow layer is decided based on the log-det term at that layer

1. The individual contribution of dimensions towards likelihood ($[L_d^{(l)}]_{s \times s \times c}$) is computed specifically for corresponding flow models (Refer Section 3.1 and Section 3.2).

2. Convert $[L_d^{(l)}]_{s \times s \times c}$ into a $\frac{s}{2} \times \frac{s}{2} \times 4c$ shaped tensor using local max and min-pooling (= −max-pooling(−input)) operations (Figure 1) at each flow layer.

3. Among the $4c$ channels, one half contains the dimensions having more log-det term compared with its neighbourhood pixel (Black marked in Fig. 1), while the other half contains the dimensions having less log-det (White marked in Fig. 1).

4. Split the tensor along the channel dimension to two parts.

5. Forward the corresponding dimensions contributing more towards likelihood into more flow layers and early gaussianize the ones contributing less.

6. Repeat steps 1-5 for all the layers with dimensions passed to that layer till the latent space.

**Training Phase:** The decision of dimensions to be factored at each layer as performed in previous step remains fixed. Finally, the flow model with proposed LCMA is trained.

---

models. Some flow models offer direct decomposition of jacobian into per-dimension components, whereas for others, an indirect estimation method has to be adopted. We now describe such methods to obtain such individual likelihood contribution of dimensions for flow models based on affine coupling (RealNVP (Dinh et al., 2016) and Glow (Kingma & Dhariwal, 2018)), and flow models involving ordinary differential equation (ODE) based density estimators (i-ResNet (Behrmann et al., 2018)), all of which involve a multiscale architecture.

### 3.1 ESTIMATION OF PER-DIMENSION LIKELIHOOD CONTRIBUTION FOR AFFINE COUPLING BASED FLOW MODELS

*RealNVP (Dinh et al., 2016):* For RealNVP with afffine coupling layers, the logarithm of individual diagonal elements of jacobian, summed over layers till layer $l$ provides the per-dimensional likelihood contribution components at layer $l$.

*Glow (Kingma & Dhariwal, 2018):* Unlike RealNVP where the log-det terms for each dimension can be expressed as log of corresponding diagonal element of jacobian, Glow contains $1 \times 1$ convolution blocks having non-diagonal log-det term for channel dimensions, for a $s \times s \times c$ tensor $\mathbf{h}$ given by,

$$\log \left| \det \left( \frac{d \, \texttt{conv2D}(\mathbf{h}; \mathbf{W})}{d \, \mathbf{h}} \right) \right| = s \cdot s \cdot \log |\det(\mathbf{W})| \tag{12}$$

It remains to decompose the $\log |\det(\mathbf{W})|$ to individual contribution by each channel. As a suitable candidate, singular values of $\mathbf{W}$ correspond to the contribution from each channel dimension, so their log value is the individual log-det contribution. So the individual log-det term for channels are obtained by,

$$|\det(\mathbf{W})| = \prod_i \sigma_i(\mathbf{W}) \Leftrightarrow \log |\det(\mathbf{W})| = \sum_i \log(\sigma_i(\mathbf{W})) \tag{13}$$

where $\sigma_i(\mathbf{W})$ are the singular values of the weight matrix $\mathbf{W}$. For affine blocks in Glow, same method as RealNVP is adopted.

### 3.2 ESTIMATION OF PER-DIMENSION LIKELIHOOD CONTRIBUTION FOR FLOW MODELS WITH ODE BASED DENSITY ESTIMATORS

Recent works on flow models such as Behrmann et al. (2018); Grathwohl et al. (2018); Chen et al. (2019) employ variants of ODE based density estimators. We introduce method to find per-dimensional likelihood contribution for i-ResNet (Behrmann et al., 2018), which is a residual network

with invertibility and efficient jacobian computation properties. i-ResNet is modelled as a flow $F(x)$, such that $z = F(x) = (I + g)(x)$, where $g(x)$ is the forward propagation function. The log-likelihood expression is written with the log-det of the jacobian is expressed as a power series,

$$\ln p_x(x) = \ln p_z(z) + \ln|\det J_F(x)|, \ln|\det J_F(x)| = \text{tr}\left(\ln\left(I + J_g(x)\right)\right) = \sum_{k=1}^{\infty}(-1)^{k+1}\frac{\text{tr}(J_g^k)}{k}$$

where $\text{tr}$ denotes the trace. Due to computational constraints, the power series is computed up to a finite number of iterations with the $\text{tr}(J_g^k)$ term stochastically approximated by hutchinson's trace estimator, $\text{tr}(A) = \mathbb{E}_{p(v)}\left[v^T A v\right]$, $\mathbb{E}[v] = 0$ and $\text{Cov}(v) = I$. The component corresponding to each dimension that becomes part of the log-det term is the diagonal element of $J_g^k$, so the per-dimension contribution to the likelihood can be approximated as the diagonal elements of $J_g^k$, summed over the power series upto a finite number of iterations $n$. The diagonal elements are obtained with the hutchinson's trace estimator without any extra cost, i.e. if $v = [v_1, v_2, \cdots, v_{s\times s\times c}]^T$,

$$\sum_{k=1}^{\infty}(-1)^{k+1}\frac{\text{tr}(J_g^k)}{k} = \sum_{k=1}^{\infty}(-1)^{k+1}\frac{\mathbb{E}_{p(v)}\left[v^T J_g^k v\right]}{k} = \sum_{k=1}^{\infty}(-1)^{k+1}\frac{\mathbb{E}_{p(v)}\left[(v^T J_g^k)v\right]}{k}$$

In above equation, $(v^T J_g^k)$ is the vector-jacobian product which is multiplied again with $v$. The individual components which are summed when $(v^T J_g^k)$ is multiplied with $v$ correspond to the diagonal terms in jacobian, over the expectation $\mathbb{E}_{p(v)}$. So those terms are the contribution by the individual dimensions, to the log-likelihood and are expressed as $[L_d^{(l)}]_{s\times s\times c}$ for use in step 1 of dimension factorization step in LCMA implementation for i-ResNet.

## 4 RELATED WORK

Multi-scale architecture and variants have been successful in a number of prior works in deep generative modeling. For invertible neural networks, Finzi et al. (2019) use a keepChannel for selective feed forward of channels analogous to multi-scaling. In the spectrum of generative flow models, multi-scale architecture has been utilized to achieve the dimensionality reduction and enhanced training because of the distribution of loss function in the network (Dinh et al., 2016; Kingma & Dhariwal, 2018). A variant of multiscale architecture has been utilized to capture local variations in auto-regressive models (Reed et al., 2017). Among GAN(Goodfellow et al., 2014) models, Denton et al. (2015) use a multiscale variant to generate images in a coarse-to-fine manner. For multi-scale architectures in generative flow models, our proposed method performs factorization of dimensions based on their likelihood contribution, which in another sense translates to determining which dimensions are important from density estimation and qualitative reconstruction point of view. Keeping this in mind, we discuss prior works on generative flow models which involve multi-scaling and/or incorporate permutation among dimensions to capture their interactions.

A number of generative flow models implement a multi-scale architecture, such as Dinh et al. (2016); Kingma & Dhariwal (2018); Atanov et al. (2019); Izmailov et al. (2019); Durkan et al. (2019); Kumar et al. (2019); Behrmann et al. (2018) etc. Kingma & Dhariwal (2018) introduce an $1 \times 1$ convolution layer in between the actnorm and affine coupling layer in their flow architecture. The $1 \times 1$ convolution is a generalization of permutation operation which ensures that each dimension can affect every other dimension. This can be interpreted as redistributing the contribution of dimensions to total likelihood among the whole space of dimensions. So Kingma & Dhariwal (2018) treat the dimensions as equiprobable for factorization in their implementation of multi-scale architecture, and split the tensor at each flow layer evenly along the channel dimension. We, on the other hand, take the next step and focus on the individuality of dimensions and their importance from the amount they contribute towards the total log-likelihood. The log-det score is available via direct/indirect decomposition of the jacobian obtained as part of computations in a flow training, so we essentially have a heuristic for free. Since our method focuses individually on the dimensions using a heuristic which is always available, it can prove to be have more versatility in being compatible with generic multi-scale architectures. Hoogeboom et al. (2019) extend the concept of $1 \times 1$ convolutions to invertible $d \times d$ convolutions, but do not discuss about multi-scaling. Dinh et al. (2016) also include a type of permutation which is equivalent to reversing the ordering of the channels, but is more restrictive and fixed. Flow models such as Behrmann et al. (2018); Grathwohl et al. (2018); Chen et al.

(2019) involve ODE based density estimators. They also implement a multi-scale architecture, but the splitting operation is a static channel wise splitting without considering importance of individual dimensions or any permutations. Izmailov et al. (2019); Durkan et al. (2019); Kumar et al. (2019); Atanov et al. (2019) use multi-scale architecture in their flow models, coherent with Dinh et al. (2016); Kingma & Dhariwal (2018), but perform the factorization of dimensions without any consideration of the individual contribution of the dimension towards the total log-likelihood. For qualitative sampling along with efficient density estimation, we also propose that factorization methods should preserve spatiality of the image in the two splits, motivated by the spatial nature of splitting methods in Kingma & Dhariwal (2018) (channel-wise splitting) and Dinh et al. (2016) (checkerboard and channel-wise splitting).

## 5 EXPERIMENTS

In Section 3, we established that our proposed likelihood contribution based factorization of dimensions can be implemented for flow models involving a multi-scale architecture, in order to improve their density estimation and qualitative performance. In this section we present the detailed results of proposed LCMA adopted for the flow model of RealNVP (Dinh et al., 2016) and quantitative comparisons with Glow (Kingma & Dhariwal, 2018) and i-ResNet (Behrmann et al., 2018). For direct comparison, all the experimental settings such as data pre-processing, optimizer parameters as well as flow architectural details (coupling layers, residual blocks) are kept the same, except that the factorization of dimensions at each flow layer is performed according to the methods described in Section 3. For ease of access, we also have summarized the experimental details in Appendix A.

For RealNVP, we perform experiments on four benchmarked image datasets: *CIFAR-10* (Krizhevsky, 2009), *Imagenet* (Russakovsky et al., 2014) (downsampled to $32 \times 32$ and $64 \times 64$), and *CelebFaces Attributes (CelebA)* (Liu et al., 2015). The scaling in LCMA is performed once for CIFAR-10, thrice for Imagenet $32 \times 32$ and 4 times for Imagenet $64 \times 64$ and CelebA. We compare LCMA with conventional RealNVP and report the quantitative and qualitative results. For Glow and i-ResNet with LCMA, we perform experiments on CIFAR-10 and present improvements over baseline bits/dim. We also perform an ablation studies for LCMA vs. other possible dimension splitting options.

### 5.1 QUANTITATIVE COMPARISON

The bits/dim scores of RealNVP with conventional multi-scale architecture (as introduced in Dinh et al. (2016)) and RealNVP with LCMA are given in Table 1. It can be observed that the density estimation results using LCMA is in all cases better in comparison to the baseline. We observed that the improvement for CelebA is relatively high as compared to natural image datasets. This observation was expected as facial features often contain high redundancy and the flow model learns to put more importance (reflected in terms of high log-det) on selected dimensions that define the facial features. Our proposed LCMA exposes such dimensions to more flow layers, making them more expressive and hence the significant improvement in code length (bits/dim) is observed. The improvement in bits/dim is less for natural image datasets because of the high variance among features defining them, which has been the challenge with image compression algorithms. Note that the improvement in density estimation is always relative to the original flow architecture (RealNVP in our case) over which we use our proposed LCMA, as we do not alter any architecture other than the dimension factorization method. The quantitative results of LCMA implementation for RealNVP, Glow and i-ResNet with CIFAR-10 dataset is summarized in Table 2. The density estimation scores for flows with LCMA outperform the same flow with conventional multi-scale architectures.

Table 1: Improvements in density estimation (in bits/dim) using proposed method for RealNVP

| Model | CelebA | CIFAR-10 | ImageNet 32x32 | ImageNet 64x64 |
|---|---|---|---|---|
| RealNVP (Dinh et al., 2016) | 3.02 | 3.49 | 4.28 | 3.98 |
| RealNVP flow model with Likelihood Contribution based Multiscale Architecture (ours) | **2.71** | **3.43** | **4.21** | **3.92** |

Table 2: Density estimation results (in bits/dim) for RealNVP, Glow and i-ResNet with LCMA trained on CIFAR-10. *Model for i-ResNet has not fully converged

| Type of multi-scale architecture | RealNVP | Glow | i-ResNet |
|---|---|---|---|
| Conventional Multi-scale Architecture | 3.49 | 3.35 | 3.45 |
| Likelihood Contribution based Multi-scale Architecture | **3.43** | **3.31** | **3.40**[*] |

## 5.2 QUALITATIVE COMPARISON

An ideal dimension factorization method should capture the local variance over series of flow layers, which helps in qualitative sampling. For LCMA implementation, we introduced local max and min pooling operations on log-det heuristic to decide which dimensions to be gaussianized early (Section 3). Figure 2(a) shows samples from original datasets, Figure 2(b) shows the samples from trained RealNVP flow model with conventional multi-scale architecture and Figure 2(c) shows the samples from RealNVP with LCMA trained on various datasets. The finer facial details such as hair styles, eye-lining and facial folds in Celeba samples generated from RealNVP with LCMA were perceptually better than the baseline. The global feature representation observed is similar to that in RealNVP, as the flow architecture was kept the same. The background for natural images such as Imagenet $32 \times 32$ and $64 \times 64$ was constructed at par with the original flow model. As has been observed in different flow models such as RealNVP and Glow, the latent space holds knowledge about the feature representation in the data. We performed linear interpolations in latent space to ensure its efficient construction and generated images, as shown in Figure 3. The interpolations observed were smooth, with intermediate samples perceptibly resembling synthetic faces, signifying the efficient construction of latent space. More interpolations are included in Appendix B.

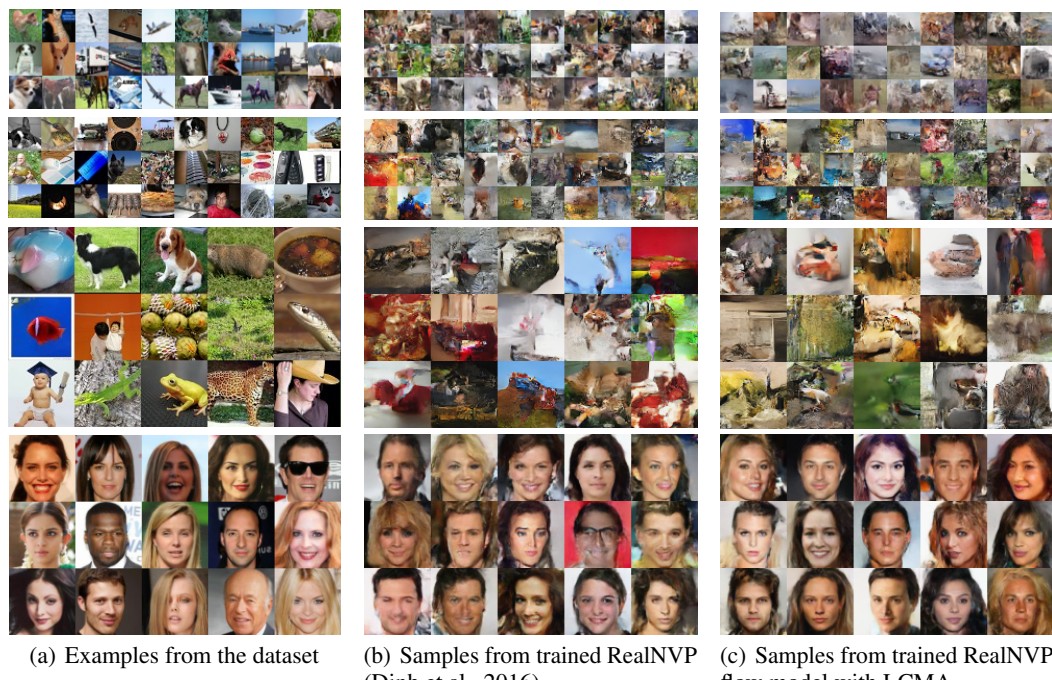

(a) Examples from the dataset    (b) Samples from trained RealNVP (Dinh et al., 2016)    (c) Samples from trained RealNVP flow model with LCMA

Figure 2: Samples from RealNVP (Dinh et al., 2016) and RealNVP flow model with proposed likelihood contribution based multiscale architecture (LCMA) trained on different datasets. The datasets shown in this figure are in order: CIFAR-10, Imagenet($32 \times 32$), Imagenet ($64 \times 64$) and CelebA (without low-temperature sampling). Additional samples are included in Appendix C.

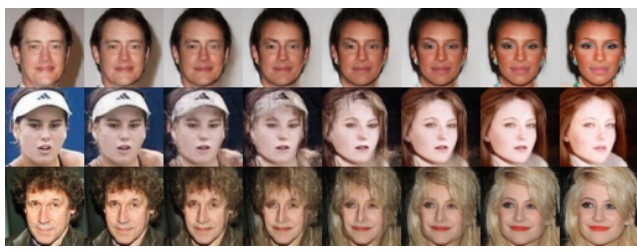

Figure 3: Smooth linear interpolations in latent space between two images from CelebA dataset (without low-temperature sampling). The intermediate samples perceptibly resemble synthetic faces.

## 5.3 ABLATION STUDY

Table 3: Ablation study results for multi-scale architectures with various factorization methods

| Evaluations | Fixed Random Permutation | Multiscale architecture with early gaussianization of *high* log-det dimensions | RealNVP (Dinh et al., 2016) | Multiscale architecture with early gaussianization of *low* log-det dimensions |
|---|---|---|---|---|
| Quantitative Evaluation (Bits/dim) | 3.05 | 3.10 | 3.02 | 2.71 |
| Qualitative Evaluation | | | | |

We performed ablation studies to compare LCMA with other methods for dimension factorization in a multi-scale architecture. We consider 4 variants for our study, namely fixed random permutation (Case 1), multiscale architecture with early gaussianization of high log-det dimensions (Case 2), factorization method with checker-board and channel splitting as introduced in RealNVP (Case 3) and multiscale architecture with early gaussianization of low log-det dimensions, which is our proposed LCMA (Case 4). In fixed random permutation, we randomly partition the tensor into two halves, with no regard to the spatiality or log-det score. In case 2, we do the reverse of LCMA, and gaussianize the high log-det variables early. The bits/dim score and generated samples for each of the method are given in Table 3. As expected from an information theoretic perspective, gaussianizing high log-det variables early provides the worst density estimation, as the model could not capture the high amount of important information. Comparing the same with fixed random permutation, the latter has better score as the probability of a high log-det variable being gaussianized early reduces to half, and it gets further reduced with RealNVP due to channel-wise and checkerboard splitting. LCMA has the best score among all methods, as the variables carrying more information are exposed to more flow layers. Fixed random permutation has the worst quality of sampled images, as the spatiality is lost during factorization. The sample quality improves for Case 2 and RealNVP. The sampled images are perceptually best for LCMA. Summarizing, LCMA outperforms multi-scale architectures based on other factorization methods, as it improves density estimation and generates qualitative samples.

## 6 CONCLUSIONS

We proposed a novel multi-scale architecture for generative flows which employs a data-dependent splitting based the individual contribution of dimensions to the total log-likelihood. Implementations of the proposed method for several state-of-the-art flow models such as RealNVP (Dinh et al., 2016), Glow(Kingma & Dhariwal, 2018) and i-ResNet(Behrmann et al., 2018) were presented. Empirical studies conducted on benchmark image datasets validate the strength of our proposed method, which improves log-likelihood scores and is able to generate qualitative samples. Ablation study results confirm the power of LCMA over other options for dimension factorization.

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

## A    EXPERIMENTAL SETTINGS

For direct comparison with Dinh et al. (2016), data pre-processing, optimizer parameters as well as flow architectural details (coupling layers, residual blocks) are kept the same, except that the factorization of dimensions at each flow layer is performed according to the method described in Section 3. In this section, for the ease of access, we summarize the experimental settings.

**Datasets:** We perform experiments on four benchmarked image datasets: *CIFAR-10* (Krizhevsky, 2009), *Imagenet* (Russakovsky et al., 2014) (downsampled to $32 \times 32$ and $64 \times 64$), and *CelebFaces Attributes (CelebA)* (Liu et al., 2015).

**Pre-processing:** For CelebA, we take a central crop of $148 \times 148$ then resize it to $64 \times 64$. For dequantization of images (whose values lies in $[0, 256]^D$), the data is transformed to $\text{logit}(\alpha + (1 - \alpha) \odot \frac{x}{256})$, where $\alpha = 0.05$. The sample allocation for training and validation were done as per the official allocation for the datasets.

**Flow model architecture:** We use affine coupling layers as introduced (Dinh et al., 2016). A layer of flow is defined as 3 coupling layers with checkerboard splits at $s \times s$ resolution, 3 coupling layers with channel splits at $s/2 \times s/2$ resolution, where $s$ is the resolution at the input of that layer. For datasets having resolution 32, we use 3 such layers and for those having resolution 64, we use 4 layers. The cascade connection of the layers is followed by 4 coupling layers with checkerboard splits at the final resolution, marking the end of flow composition. For CIFAR-10, each coupling layer uses 8 residual blocks. Other datasets having images of size $32 \times 32$ use 4 residual blocks whereas $64 \times 64$ ones use 2 residual blocks. More details on architectures will be given in a source code release.

**Optimization parameters**: We optimize with ADAM optimizer (Kingma & Ba, 2014) with default hyperparameters and use an $L_2$ regularization on the weight scale parameters with coefficient $5 \cdot 10^{-5}$. A batch size of 64 was used. The computations were performed in NVIDIA Tesla V100 GPUs.

**Multiscale Architecture:** Scaling is done once for CIFAR-10, thrice for Imagenet $32 \times 32$ and 4 times for Imagenet $64 \times 64$ and CelebA.

## B    INTERPOLATIONS AMONG TWO IMAGES FROM CELEBA DATASET

Figure 4 presents more interpolation examples obtained using our model between two images from CelebA dataset.

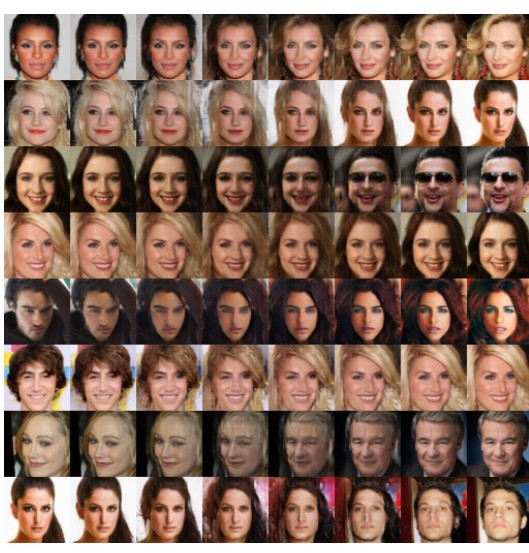

Figure 4: Linear interpolations between two CelebA images (without low-temperature sampling)

## C  ADDITIONAL SAMPLES

In this section, we present more samples from RealNVP model with likelihood contribution based multiscale architecture trained on different datasets.

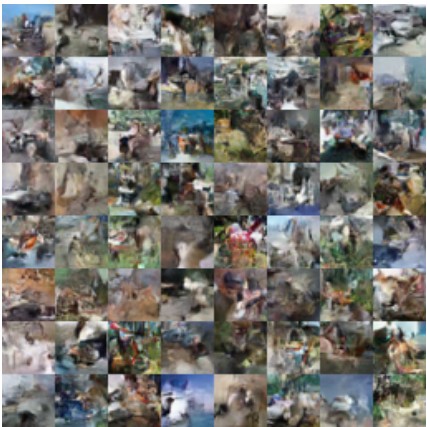

Figure 5: Samples from model trained on CIFAR-10 dataset

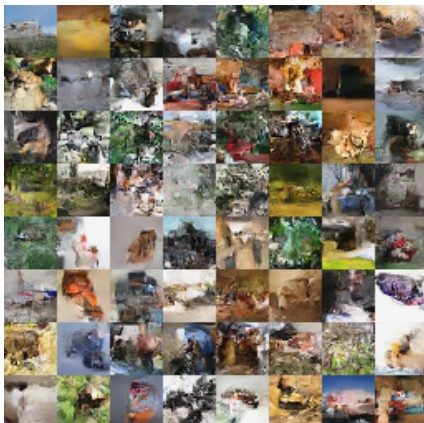

Figure 6: Samples from model trained on Imagenet $32 \times 32$ dataset

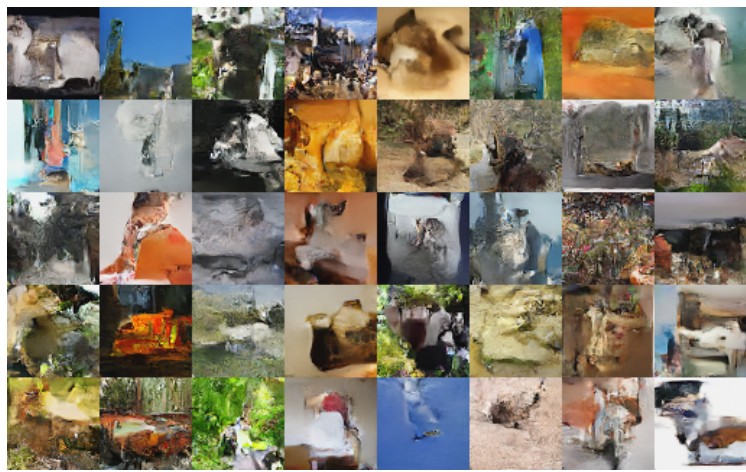

Figure 7: Samples from model trained on Imagenet $64 \times 64$ dataset

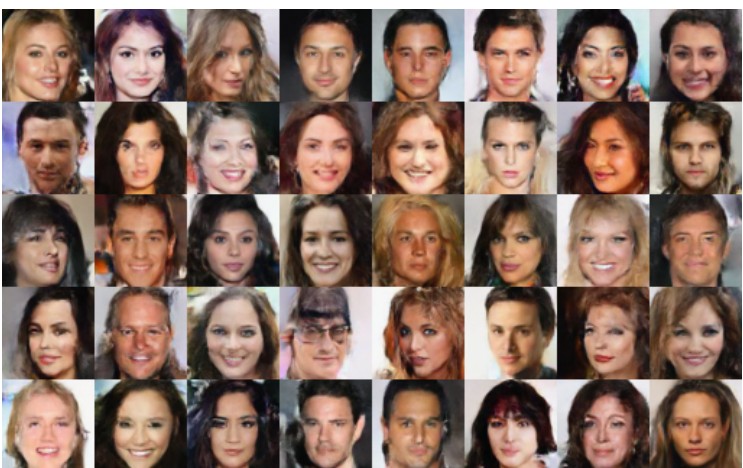

Figure 8: Samples from model trained on CelebA dataset without low-temperature sampling

