# OpenReview forum: "Likelihood Contribution based Multi-scale Architecture for Generative Flows"
_ICLR.cc/2020/Conference — Reject_

### Official Review · AnonReviewer2 · 2019-10-23
**Official Blind Review #2**

**Rating:** 3

**Review:**

Summary:
The paper investigates a strategy for factoring out dimensions in multi-scale generative flow models. The strategy relies on a heuristic that employs the availability of per-dimension log-determinant terms (e.g. as affine couplings) to decide which half of the dimensions to factor out. Quantitative experimental results support the proposed strategy’s efficacy.

Recommendation: Weak Reject
The paper misses several important comparisons and baselines - both in terms of placing the proposed method in a broader context (e.g. how could the method be used in multiscale i-ResNet), as well as providing experimental comparisons that would definitively demonstrate the method’s contribution (e.g. is it the heuristic dimension splitting or the unfactored pre-training that provides the benefit.

The applicability of the proposed method appears to be overstated. The paper suggests that the method is applicable to any multi-scale flow architecture. However, it depends on the availability of per-dimension log-determinants. As the paper states, these values are readily available for methods that rely on affine couplings (e.g. RealNVP, Glow), but not for methods such as i-ResNets or FFJORD, that also employ multi-scale architectures to factor out variables.


Major comments:
1. Section 2.1 introduces flows as a mapping from data x to the latent code z, and gives the log-likelihood formula (2) that can be optimized during model training. This presentation is at odds with Section 3, where the authors talk about sampling a latent variable that fixes the “log-latent density” and only leaves the log-determinant term to be optimized (e.g. “[...] the log-latent-density term depends on the choice of the latent distribution and if fixed given the sampled latent variable. [...] once the variable is sampled, maximizing the log-det term results in maximized likelihood.” The two presentations are at odds with each other and should be reconciled, ideally by altering Section 3 since it does not reflect the way flow models are trained.
2. The authors made an implicit assumption that the log-det term is decomposable into per-dimension component. As mentioned earlier in this review, generally speaking this is not true and methods such as i-ResNets or FFJORD do not (directly?) permit such decomposition, while they do employ multi-scale architectures akin to that of RealNVP. This assumption manifests itself in several places in the paper, which are listed below. Either the authors could make this assumption and the limitations imposed by it very clear, or they could provide additional information on how their method extends to methods that do not provide per-dimension log-det contributions.
a) In Section 3 “The total log-det term is nothing by the sum of log-det terms for contributed by each dimension”. This is obvious (and possibly holds) only for affine couplings.
b) In Section 3 “Nevertheless, such an analogy can be extended for other flow models which involve a multiscale architecture [...]”. As mentioned, I am not convinced this is true.
c) In Section 4 “Since our method focuses individually on the dimensions using a heuristic which is always available in flow model training, it can prove to be have more versatility in being compatible with generic multi-scale architecture”. Same as above - in fact I believe that the opposite may be true.
d) Section 4 does not mention methods that employ multi-scale architectures whil not allowing for discerning per-dimension contributions of the log-det. Mentioning these methods is important for accurately positioning the authors contribution relative to the existing literature.
e) The method is only applied to the RealNVP architecture. The claim of it being generally applicable to multi-scale architectures would be much stronger if the comparison would include other multi-scale architectures. Here, I strongly recommend i-ResNets or a similar architecture based on log-det approximation to the entire coupling layer.
3. Unless I missed it, the authors do not mention how they aggregate the log-det values. They are available per datapoint. Are they aggregated over the entire dataset?
4. In Section 4 the authors imply that the contribution of a dimension towards the total likelihood is dominated by the log determinant (“[...] as the contribution by the variable (dimension) towards the total log-det (~ total log-likelihood)”). This is not obvious, and I am not sure it is always true - it probably depends on the choice of the base distribution p(z).
5. The authors show experimentally that the use of their variable factorisation heuristic improves log-likelihoods over the baseline RealNVP model; they further perform ablation studies to confirm that preferential factorisation of low log-det dimensions provides best log-likelihoods and qualitative results on the CelebA dataset. However, several points remain unclear.
a) What are the relative contributions of pre-training without dimension factorisation and the proposed factorisation heuristic? Do the experiments shown in Table 2 all employ the pre-training scheme? If so, this should be clearly stated. If not, this set of experiments should be performed.
b) The networks used in the experiments use between m=1 and 4 factorisation layers. So in total, there are 2^m possible decisions of which half of the variables at those layers to factorise. These numbers are not so large as to prevent us from training all possible combinations of the networks. How does the proposed heuristic compare to just training all (or a few sampled) factorisations decided a priori? This comparison should also investigate the effect of using pre-training.
6. In the qualitative comparison (Section 5.2) the authors argue two points (i) RealNVP with the proposed method generates finer details on CelebA; and (ii) provides a reasonable latent space that can be interpolated in.
a) With regards to (i), I personally find it difficult to argue about fine details in what are effectively low-resolution images. The differences (which are already hard to see and easy to imagine) could be present due to minute changes in the training code/procedure/random seed/etc. I would be more confident about these differences if the authors could confirm that the methods were trained using the same code base, random seeds, dataset shuffle and framework versions.
b) How do the qualitative interpolation results compare to RealNVP without the proposed factorisation heuristic?

Minor comments:
- In Section 2.1 “Let x be a high-dimensional random vector with unknown true distribution x ~ p(x) [...]”. Here x appears to refer to the data or a sample from the data distribution, and not just a random vector. This could be made clearer.
- In Section 2.1 the “jacobian” should be written with a capital “J”
- In Section 2.1 and elsewhere, when denoting outputs of intermediate flows f_i as y_i, it is mentioned that y_0 = x, but not that y_k = z. If the latter is true, mentioning this would improve clarity.
- Section 3 “In a multi-scale architecture, it is apparent that the variables getting exposed to more layers of flow will be more expressive in nature [...]”. I do not find this to be an obvious fact. Perhaps the authors could provide an intuition for why this is true or demonstrate this on a toy example? My confusion may stem from the fact that I am unsure what it means for a variable (i.e. a dimension we factor out) to be more expressive.
- Some phrasing is a little unusual (e.g. “less (more) log-det tems” instead of “smaller (larger) log-det term”; “to be have”).
- In Section 5.1 “code length” is introduced without context. The connection between code length, log-likelihood and compression can be made more clear.

------------------------------------------------
Post rebuttal update
------------------------------------------------

I’d like to thank the authors for additional elaboration on their baseline experiments that consider their heuristic-based scheme for factoring out dimensions (the low-low-low scheme) and the “anti-heuristic” (which considers the high-high-high scheme). I agree with the authors that both schemes would have been within the 2^m possibilities I encourage them to consider, however I do not fully agree with some of the conclusions the authors draw for these experiments.

First, given that the method proposed by the authors is a heuristic, it is not obvious that the high-high-high and the low-low-low schemes indeed correspond the worst and best possible of the 2^m schemes. Running the 2^m experiments could demonstrate this empirically and strengthen the manuscript.

Second, if we speak about the high-high-high and low-low-low factorizations, then we’ve already gone through the computationally expensive effort of pre-training a model that does not factor out any variables. It is precisely because of this that I’d like to see a baseline that doesn’t do any pre-training, but simply enumerates the 2^m schemes and selects the one with the best validation score. It is not obvious that this approach would computationally more expensive than the pre-training step -- another point I believe is important to cover.

To summarize, I do not believe that the baselines currently present in the paper are a substitute for the ones requested.
Including new results requires a review of those new results, so I would not be in favour of adding new results at the camera ready stage. I therefore retain my score.


**Experience Assessment:**

I have published in this field for several years.

**Review Assessment: Checking Correctness Of Derivations And Theory:**

I assessed the sensibility of the derivations and theory.

**Review Assessment: Checking Correctness Of Experiments:**

I carefully checked the experiments.

**Review Assessment: Thoroughness In Paper Reading:**

I read the paper at least twice and used my best judgement in assessing the paper.

---

> ### Author Response · Authors · 2019-11-13
> **Author response part 1**
>
> Thank you for providing a detailed feedback on our submission and raising interesting comments. Please see our detailed responses below.
>
> Response to Major Comments:
> 1. Thank you for pointing this out. We stick to the notation introduced in section 2.1 and have modified section 3 to be coherent with the standard method for flow training.
>
> 2. It has been correctly pointed out that some flows such as affine coupling based flows readily provide the per-dimension log-det components. But since the flow training always has jacobian term which encompasses contribution from all the dimensions, a method, direct/indirect can be designed to estimate the per-dimensional contribution towards the log-likelihood. The class of flow models which involve variants of ordinary differential equation (ODE) based density estimators such as i-ResNet (Behrmann et al., 2018), FFJORD (Grathwohl et al., 2018) and residual flows (Chen et al., 2019) approximate the log-determinant term by a power series of trace of jacobians which is estimated by hutchinson's trace estimator. For this class of flow models, the contribution coming from each dimension towards the log-likelihood can be approximated as the component that becomes part of the trace in power series sum, i.e. the diagonal elements of the jacobian. The terms can be readily obtained during the trace estimation step.
>
> We noticed this to be a common comment across all the reviewers. So, we implemented likelihood contribution based multi-scale architecture (LCMA) for Glow (Kingma and Dhariwal, 2018) and i-ResNet. The implementation details are included in Section 3.1 and 3.2 and the density estimation results for CIFAR-10 are included in Table 2 in the updated manuscript. please refer to the author comment "Additional experiments as per common comments from the reviewers and updated manuscript" addressing the common comment by all the reviewers for a brief description.
>
> The sub-comments 2(a,b,d,e) are addressed in above responses. For 2(c), we meant to mention that flow training always includes log-det of jacobian term and hence the heuristic can always be obtained by direct/indirect decomposition of the log-det term to per-dimensional components. We have updated the corresponding lines in Section 4 to incorporate this comment.
>
> 3. The log-det values at each layer are aggregated and averaged over the entire dataset.
>
> 4. We believe this comment pertains to Section 3 and not Section 4. The likelihood indeed depend on the choice of base distribution, but the modelling of flow layers (with the choice of dimensions splitting method in a multi-scale architecture) is reflected in the log-det component. So, we aim to efficiently design the flow layers based on the log-det score to maximize the total likelihood.
>
> 5. We believe there is a misunderstanding here about the training process with LCMA. The pre-training is performed without any multiscaling for the sole purpose of obtaining the log-det value for every dimension at each flow layer. That is the only result we utilize from the pre-training process for deciding which dimensions to be factored out early. Once the decision is made, the final training with the multi-scale architecture is performed independently to that of pre-training.
>
>
> (a) In Table 3 in updated manuscript (Table 2 in old manuscript) which presents the ablation study results, the Multiscale architecture with early gaussianization of high log-det dimensions and with early gaussianization of low log-det dimensions include pre-training since for both these cases, the decision of dimensions to be factored out depends on the log-det score obtained during pre-training. For fixed random permutation and RealNVP, the decision for dimension factorization is static and non-data-dependent, so pre-training is not required for them.
>
> (b) Our ablation study shows that early gaussianization of high log-det dimensions has a deteriorating effect on the density estimation score. This implies the dimensions corresponding to high log-det carry more information. So, we just consider the case where high log-det dimensions are put into more flow layers. However training all $2^m$ combinations can be an interesting step to understand more deeply into the interplay of dimensions.
>
> 6. (a) We confirm that for all the flow models, all the experimental details were kept intact except changing the dimension splitting mechanism at each flow layer from static to data-dependent via likelihood contribution.
>
> (b) The aim of the interpolations was to ensure efficient construction of latent space, so we do not compare nor report any improvements in interpolations over the original flow model.

---

> > ### Author Response · Authors · 2019-11-13
> > **Author response part 2**
> >
> > Response to Minor Comments:
> > We have updated our manuscript for minor comment no. 1,2,5,6. For minor comment 3, in section 2.1 we don't mention $y_k = z$ as it does not hold true for multi-scale architecture (for which equations 6-8 hold true). But in section 3 when we assume no multi-scaling, we explicitly mention $y_k = z$. For minor comment 4, when we mention dimensions to be more expressive, we mean the information about the distribution for that dimension is well captured by the flow. In LCMA, we pass high log-det dimensions into more flow layers so that the flow learns to better capture information about them. We have updated the corresponding line in section 3.
> >
> > Glow: D. P Kingma and P. Dhariwal. Glow: Generative flow with invertible 1x1 convolutions, In Advances in Neural Information Processing Systems, pp. 10215–10224, 2018.
> > i-ResNet: J. Behrmann, W. Grathwohl, R.TQ Chen, D. Duvenaud, and J. Jacobsen, Invertible residual networks.arXiv preprint arXiv:1811.00995, 2018
> > FFJORD: W. Grathwohl, R. TQ Chen, J. Betterncourt, I. Sutskever, and D. Duvenaud, Ffjord:Free-form  continuous  dynamics  for  scalable  reversible  generative  models.arXiv  preprintarXiv:1810.01367, 2018
> > Residual Flows: R. TQ Chen, J. Behrmann, D. Duvenaud, and J. Jacobsen, Residual flows for invertible generative modeling.arXiv preprint arXiv:1906.02735, 2019
> >
> > Hope this addresses all your comments.

---

> > > ### Comment · AnonReviewer2 · 2019-11-15
> > > **Response to rebuttal**
> > >
> > > I would like to complement the authors on the additional experiments they performed w.r.t. applying their heuristic to the i-ResNet flow architecture, which does not readily provide per-dimension estimates of the log-determinant contributions. These additional results, and the accompanying methods (Section 3.2) demonstrate that the method is more generally applicable and thus substantially strengthen the paper. Furthermore, I’d like to thank the authors for clarifying details of their method in their response (eps. w.r.t. to pre-training).
> > >
> > > However, I still feel that an important baseline is missing. Specifically, I believe that demonstrating that the proposed heuristic outperforms the naive (and very simple to implement) approach of considering all 2^m  factorisation outperforms (either in expended computational budget, or in log-likelihood) is critical to assessing the value of the author’s contribution.
> > >
> > > While I believe that the paper has been substantially improved during the rebuttal process, I am not yet prepared to increase my recommendation to a “Weak Accept” without the additional results mentioned above.

---

> > > > ### Author Response · Authors · 2019-11-15
> > > > **We already have performed related ablation experiments**
> > > >
> > > > Thank you for your kind words and complementing our effort.
> > > >
> > > > We wanted to state that we already have related experiments in our ablation study section. For a flow with lets say having 4 layers in total, and 3 of them where dimension splitting is performed, at each layer the low or high log-det dimensions can be early gaussianized, giving $2^3$ possibilities such as high-high-high,...,low-low-low. The "Multiscale architecture with early gaussianization of high log-det dimensions" in our ablation study corresponds to high-high-high and "Multiscale architecture with early gaussianization of low log-det dimensions" or LCMA corresponds to low-low-low. The ablation study confirms that early gaussianizing high log-det dimensions (high-high-high) gives worst density estimation performance as compared to the optimal choice (low-low-low), which implicitly implies that other combinations with a "high" such as high-low-high or high-high-low will underperform as compared to low-low-low. Having explicit results for all the combinations requires computation time, and we will be happy to include the results in the final version, if accepted.

---

### Official Review · AnonReviewer1 · 2019-10-27
**Official Blind Review #1**

**Rating:** 3

**Review:**

This paper propose a heuristic algorithm for deciding which random variables to be Gaussianized early in flow-based generative models. The proposed algorithm involves first training a flow without multi-scale training, for example, 32*32*c  - 32*32*c - 32*32*c. Then, it computes the logdet term for each variable at each layer. It then spatially partition the first flow block by two halves of shape 16*16*2c based on max-pooling the logdet term. Then it recursively Gaussianize one half, and partition the other half as 8*8*4c, still using the pre-computed logdet tensors (Ld in the paper). After partitioning, they train a multi-scale model with the learned partition.

While I agree adaptive multi-scale architecture is a topic worth researching, and the paper does have some positive experimental results. I think the writing of the paper is very vague and the techniques are not sensible.

Writing: the main algorithm is just depicted in the last paragraph of Page 4. There are not any equations or pseudocode on what exactly does the proposed algorithm do. Figure 1 and 2 are not detailed enough. For example, Figure 2 doesn't explain how to "Perform splitting based on log-det heuristic and spatial constraints". I can only guess what the algorithm is. I suggest the authors make the algorithm more clear, and avoid using large paragraphs of natural language to depict the algorithm.

Technique:
1. While I agree partitioning based on logdet term makes some sense, I think *recursively* partitioning without updating the logdet terms is problematic. If the flow only have one layer, the proposed algorithm makes sense. However, for a multi-layer flow model. After the first partitioning, the network changes. For example, for a two layer flow 32*32*c - 32*32*c - 32*32*c, the two layers both have 3*3*c*c filters. However, after partitioning the first output layer as two 16*16*2c, the filter of the second layer should have shape 3*3*2c*2c now. It is not clear how to translate the original, single-scale model into a multi-scale one. And the logdet tensor for the second flow layer doesn't mean anything now.

2. For affine coupling layer, we can indeed compute the logdet term dimension wise. However, it is not clear how to do the computation for other types of flows. For example, invertible ResNets, which estimates the log-det term for each ResBlock with an unbiased estimator.

3. I am also not sure with the training algorithm after pretraining. Do we need to remember what pixel to pickup for each max-pooling operation? That has O(s*s*c*L) space complexity. Does the "gather" operation baesd on the memorized locations time consuming?

4. Training a single-scale model has higher time complexity than training a multi-scale model. Is this time complexity too high?

=================

Update: thank for the authors for their significant effort on revising this paper.

Writing is indeed much better. However there are still many typos (e.g. jabocian, algorithm). Algorithm 1 is better than the original plain-text version, it still doesn't look like even a pseudocode though. I suggest converting the bullets into actual code, e.g., (try to minimize the amount of natural language since it is vague)

For each layer l
Ld1, Ld2 <- Maxpooling(sth), Minpooling(sth)
...
EndFor

I still don't think my concern 1 is addressed. Imagine a single-scale flow

y = AffineCoupling(x)
z = AffineCoupling(y)
h = AffineCoupling(z)

vs a multi-scale flow

y = AffineCoupling(x)
y1, y2 = LCMA-Split(y)
z = AffineCoupling(y1)
z1, z2 = LCMA-Split(z)
h = AffineCoupling(z1)

I don't think |dh/dz1| of model 2 is a submatrix of |dh/dz| of model 1. Because z1 of model 2 is not a part of z of model 1 in the first place. In model 2, z1 is computed with only y1, while in model 1, z1 is computed with the full y.

Concern 3:
The authors partially addressed my question. However, the proposed algorithm is still more expensive than a fixed multi-scale architecture, right? A fixed multi-scale architecture such as RealNVP (next scale block is s/2 * s/2 * 2c) is cheaper than a single-scale architecture (next scale block is s/2 * s/2 * 4c). I guess the time complexity of the proposed approach is the letter one instead of the first one. So the improved likelihood still comes with time complexity cost (comparing with a fixed multi-scale architecture).

To summarize, I can increase my score to 3 as a positive feedback for the author's effort. But I really think this paper still has a long way to go to be complete.

**Experience Assessment:**

I have read many papers in this area.

**Review Assessment: Checking Correctness Of Derivations And Theory:**

I assessed the sensibility of the derivations and theory.

**Review Assessment: Checking Correctness Of Experiments:**

I assessed the sensibility of the experiments.

**Review Assessment: Thoroughness In Paper Reading:**

I read the paper at least twice and used my best judgement in assessing the paper.

---

> ### Author Response · Authors · 2019-11-13
> **Author Response**
>
> Thank you for providing constructive comments on our submission and resonating with the importance of adaptive multi-scale architecture as an interesting research area. Please see our detailed responses below.
>
> Response to comment on writing: Thank you for the suggestion to remodel the writing of the algorithms section. We have updated the manuscript with vivid explanation of the likelihood contribution based dimension splitting algorithm for multi-scale architecture. We have replaced original figure 2 with detailed description of different phases of the training process for flow model with likelihood contribution based multiscale architecture (LCMA), of which the data dependent dimension splitting sits at the core.
>
> Response to comments on technique:
> 1. For multi-layer flow models, we monitor the dimensions which get gaussianized early and which pass through more layers at every flow layer. Since the squeezing operation which converts $s\times s\times c$ tensor to a $\frac{s}{2}\times \frac{s}{2}\times 4c$ tensor is nothing but reordering of dimensions, we can easily monitor which dimensions belong to the half ($\frac{s}{2}\times \frac{s}{2}\times 2c$) that get gaussianized early and which dimensions belong to the other half that passes through more flow layers. At the next layer, only the log-det terms for the dimensions which came through flow layers are considered for further splitting operation.
>
> 2. This is a comment that we commonly received from all the reviewers. We have implemented LCMA for Glow and i-ResNets and have updated the manuscript with the implementation details (Section 3.1 and 3.2) and density estimation results for CIFAR-10 (Table 2). Please refer to the author comment "Additional experiments as per common comments from the reviewers and updated manuscript" addressing the common comment by all the reviewers for a brief description. The same analogy as i-ResNets can be extended for a number of other flows with ODE based density estimators such as FFJORD (Grathwohl et al. 2018) and residual flows (Chen et al. 2019).
>
> 3. and 4. The max and min pooling operations are not performed online during the training process. As we describe in "Dimension Factorization" phase in Algorithm 1 in the manuscript, the max and min pooling operations are performed inbetween the pre-training and final training just to decide at each layer which dimensions to gaussianize early and which ones to pass through more flow layers. We design masks at each flow layer based on above decision, which are used during the final multi-scale flow training, which starts after the decision is made. Our data-dependent dimension splitting operation can also be interpreted as replacing the conventional checkerboard/channel split masking with likelihood contribution based masking. So to summarize, the time complexity associated with original multi-scale architecture is not changed as we dont add any additional time consuming blocks.
>
> Glow: D. P Kingma and P. Dhariwal. Glow: Generative flow with invertible 1x1 convolutions, In Advances in Neural Information Processing Systems, pp. 10215–10224, 2018.
> i-ResNet: J. Behrmann, W. Grathwohl, R.TQ Chen, D. Duvenaud, and J. Jacobsen, Invertible residual networks.arXiv preprint arXiv:1811.00995, 2018
> FFJORD: W. Grathwohl, R. TQ Chen, J. Betterncourt, I. Sutskever, and D. Duvenaud, Ffjord:Free-form  continuous  dynamics  for  scalable  reversible  generative  models.arXiv  preprintarXiv:1810.01367, 2018
> Residual Flows: R. TQ Chen, J. Behrmann, D. Duvenaud, and J. Jacobsen, Residual flows for invertible generative modeling.arXiv preprint arXiv:1906.02735, 2019
>
> Hope this addresses all your comments.

---

### Official Review · AnonReviewer4 · 2019-11-04
**Official Blind Review #4**

**Rating:** 3

**Review:**

This paper presents a new multi-scale architecture for flow-based generative models. Unlike prior work on multi-scale flow architectures which use fixed dimension-splitting heuristics, the proposed approach learns which dimensions to process further. The features are chosen for further processing based on a heuristic motivated by each feature's contribution to the total likelihood. This contribution is given by the each features' contribution to the log-determinant term in the change of variables formula. The model is trained in a two-step process. First a flow model with no multi-scale architecture is trained. Then each feature's  importance is calculated based on its contribution to the log-determinant. Then these scores are used to rank the features for the second, multi-scale model which is retrained from scratch. The authors demonstrate the performance of their approach on density modeling on standard image datasets. They demonstrate an improvement over the standard real-nvp architecture.

Overall, I would recommend to reject this paper. While the proposed method is interesting, I do not feel that the authors  experiments demonstrated its performance relative to other additions on top of coupling-layer based flow models. For example, one could view the invertible 1x1 convolutions from GLOW as a similar way to "``learn" how the features should be ordered. The authors do not provide a comparison with more modern flows such as Glow, Residual Flows -- both of which perform considerably better than Real-NVP with the proposed architecture.

While I think it is possible that Glow, Residual Flows and other recent flows could benefit from the proposed architecture, the results presented here do not convince me of this fact. If the authors presented a more comprehensive ablation study comparing their architecture to the various additions of other recent work on large-scale flow-based generative models, and their model performed favorably, then I could be convinced to raise my score.

__________________
POST REBUTTAL
__________________

I appreciate the great efforts the authors have gone through to address my initial concerns. Most of these concerns had to do with the limited scope of the paper's initial experiments. The authors have expanded their experiments to incorporate a few more flow-based generative models and have demonstrated their proposed approach can improve their likelihoods across the board. I am somewhat disappointed that they did not attempt to improve upon the current SOTA on CIFAR10 (residual flow) despite the fact that code is available for this model. If the authors had used their method to improve upon the SOTA then their experiments would have been considerably more convincing.

Despite that, I find these new results much more compelling and I would like to make that clear to the area chairs. Given the harsh discretization of the scoring system (1, 3, 6, 8) I cannot in good faith increase my score from a 3 to a 6, but I would like to make it very clear that these changes to the experiments do improve my view of the paper -- moving from a weak reject to neutral, though that is a not an official option.


**Experience Assessment:**

I have published in this field for several years.

**Review Assessment: Checking Correctness Of Derivations And Theory:**

I did not assess the derivations or theory.

**Review Assessment: Checking Correctness Of Experiments:**

I assessed the sensibility of the experiments.

**Review Assessment: Thoroughness In Paper Reading:**

I made a quick assessment of this paper.

---

> ### Author Response · Authors · 2019-11-13
> **Included comparison with $1\times 1$ convolutions and LCMA implementations for Glow and i-ResNet**
>
> We thank the reviewer for their constructive suggestions and are pleased that the reviewer acknowledges our proposed method as interesting. Please see our detailed responses below.
>
> Reviewer: "For example, one could view the invertible 1x1 convolutions from GLOW as a similar way to "``learn" how the features should be ordered."
>
> Response: We discussed the comparison with Glow (Kingma and Dhariwal, 2018) in "Related Work" section. The $1\times 1$ convolution is a generalization of permutation operation which can be interpreted as a step towards redistributing the contribution of dimensions to total likelihood among the whole space of dimensions. For this reason, authors in Glow treat the dimensions as equiprobable for factorization in the implementation of multi-scale architecture, and split the tensor at each flow layer evenly along the channel dimension. We, on the other hand, take the next step and focus on the individuality of dimensions and their importance from the amount they contribute towards the total log-likelihood, which can be implemented on top of $1\times 1$ convolutions.
>
> We implemented likelihood contribution based multi-scale architecture (LCMA) for Glow, and tested it for CIFAR-10 dataset. LCMA implementation for Glow leads to improved density estimation. We have added Section 3.1 with implementation details and Table 2 with density estimation results in the updated paper.
>
> Reviewer: "The authors do not provide a comparison with more modern flows such as Glow, Residual Flows -- both of which perform considerably better than Real-NVP with the proposed architecture." and
> "If the authors presented a more comprehensive ablation study comparing their architecture to the various additions of other recent work on large-scale flow-based generative models, and their model performed favorably, then I could be convinced to raise my score."
>
> Response: We noted this comment to be common across all the reviewers. So we conducted additional experiments by implementing likelihood contribution based multiscale architecture for Glow and i-ResNet (Behrmann et al., 2018) and tested them for CIFAR-10, please refer to the author comment "Additional experiments as per common comments from the reviewers and updated manuscript" addressing the common comment by all the reviewers for a brief description.
>
> The LCMA implementation details are included in Section 3.1 for RealNVP and Glow and in Section 3.2 for i-ResNet in the updated paper. The density estimation results for CIFAR-10 are included in Table 2. The bits/dim score for CIFAR-10 improves with LCMA as compared to conventional multi-scale architecture which involves static methods for dimension splitting. Please note that flow models such as i-ResNet, FFJORD (Grathwohl et al., 2018)  and residual flows (Chen et al., 2019) all employ variants of ODE based density estimators where the log-det term is estimated as a trace with hutchinson's trace estimator. We picked i-ResNet as a representative of the class of ODE density estimator based generative flows and implemented LCMA for it.
>
> Glow: D. P Kingma and P. Dhariwal. Glow: Generative flow with invertible 1x1 convolutions, In Advances in Neural Information Processing Systems, pp. 10215–10224, 2018.
> i-ResNet: J. Behrmann, W. Grathwohl, R.TQ Chen, D. Duvenaud, and J. Jacobsen, Invertible residual networks.arXiv preprint arXiv:1811.00995, 2018
> FFJORD: W. Grathwohl, R. TQ Chen, J. Betterncourt, I. Sutskever, and D. Duvenaud, Ffjord:Free-form  continuous  dynamics  for  scalable  reversible  generative  models.arXiv  preprintarXiv:1810.01367, 2018
> Residual Flows: R. TQ Chen, J. Behrmann, D. Duvenaud, and J. Jacobsen, Residual flows for invertible generative modeling.arXiv preprint arXiv:1906.02735, 2019
>
> Hope this addresses all your comments.

---

> ### Author Response · Authors · 2019-11-14
> **We will include LCMA implementation and result for residual flows in final version of the manuscript**
>
> Thank you for appreciating our efforts in the rebuttal process.
>
> We received a common comment about potential of LCMA being applicable for flow models with ODE based estimators, and majority of the reviewers explicitly mentioned about the applicability for i-ResNet. Since all the models in the family of flow models with ODE based estimators such as i-ResNet, FFJORD and residual flows employ similar methods for log-det estimation, LCMA implementation for them will be similar. Given the limited amount of time available in rebuttal period, we decided to pick i-ResNet as the representative flow for the above family and implemented LCMA for the same, along with Glow, as per your suggestion.
>
> We will implement LCMA for residual flows and include the result in the final version of the manuscript, if accepted.

---

### Public Comment · ~Yilun_Xu1 · 2019-10-24
**Questions about the equation (9)**

Hi! I am confused about the equation (9), since the LHS term is a s*s*c tensor, but the RHS is a scalar. Could you help me to understand this equation? Thank you very much.

---

> ### Author Response · Authors · 2019-10-25
> **Both LHS and RHS will have the same dimension**
>
> Hi Yilun,
> Thanks for your interest.
> Both LHS and RHS in equation 9 will have the same dimension.
>
> Assuming no dimensionality reduction at any flow layer and the dimension at intermediate layer $l$ being $s\times s\times c$,
>
> LHS: Log-det term is expressed as a $s\times s\times c$ tensor in which each term corresponds to contribution by the corresponding dimension.
> RHS:  The RHS is expressed as a $s\times s\times c$ array. Recalling that the explicit jacobian expression for affine flows (eq. 6 in RealNVP, https://arxiv.org/pdf/1605.08803.pdf), the determinant of the jacobian is product of all the diagonal elements (which will be $s\times s\times c$ in total, same as number of variables). So instead of taking the pooled product, we retain the $s\times s\times c$ structure where the terms correspond to the diagonal terms of the jacobian expression (eq.6, RealNVP). The total log-det for that component flow is nothing but sum of individual logarithms of the $s\times s\times c$ array elements. We accumulate the log-det contribution from each layer to have cumulative log-det contribution at layer $l$, in coherence with how the total log-det is calculated while training.
>
> Please note that if there is squeezing operation involved in a particular flow layer (without dimensionality reduction), which is nothing but reordering of dimensions, both LHS and RHS terms in eq. 9 are also squeezed to match the dimension of variables at that layer. This helps in correctly accumulating the contribution by each dimension to total log-likelihood across the layers by keeping track of dimensions after each squeezing operation.
>
> Hope this helps you in understanding.

---

> > ### Public Comment · ~Yilun_Xu1 · 2019-10-27
> > **Still don't quite get it**
> >
> > Hi!
> >
> > Thank you for your patient explanation! I still don't understand why the LHS is a tensor. Since det(matrix) is a scalar after all. In other word, could you please further explain the sentence "So instead of taking the pooled product, we retain the  structure where the terms correspond to the diagonal terms of the jacobian expression (eq.6, RealNVP)"? ( My guess: you retain the diagnoal elements of  jacobian of the i-th layer , denote the matrix as diag(J_i). Then you sum up all the diag(J_i). In this case, I don't recommend to experss it in the form like Eq (9).)
> >
> >
> > Thank you very much!
> >
> > Yilun

---

> > > ### Author Response · Authors · 2019-10-30
> > > **Your understanding is correct**
> > >
> > > Hi Yilun,
> > > Your understanding is correct, we retain the diagonal elements of the jacobian at $i^{th}$ layer, except instead of writing it as diag($J_i$), we express it in $s\times s\times c$ shape. For the clarity of other readers, let us explain the concept with example of a $2\times 2\times 1$ image ($\begin{bmatrix}pix_1&pix_2\\pix_3&pix_4\end{bmatrix}$). Since there are 4 variables, the jacobian at layer $i$ will be of shape $4\times 4$ which as per eq.6 of [RealNVP] is,
> > > $$\frac{\partial y_i}{\partial y_{i-1}^T} = \left[\begin{array}{cc}
> > > \mathbb{I}_{2\times 2} & 0 \\
> > > \frac{\partial y_{i(3,4)}}{\partial y_{i-1(1,2)}^T} & \big(\text{diag}\big(\exp\left[s\left(y_{i-1(1,2)}\right)\right]\big)\big)_{2\times 2}
> > > \end{array} \right]$$
> > > Which is assuming the top row variables (corresponding to $pix_1$ and $pix_2$) are passed without change and bottom row variables (corresponding to $pix_3$ and $pix_4$) are expressed as affine sum as per eq. 7 of [RealNVP].
> > > Now, the log-det of jacobian can be expressed in two ways:
> > > Expression 1:
> > > $[\text{log det of jacobian}]_{scaler} = log\big(1 \times 1 \times \exp\left[s\left(y_{i-1(1,2)}\right)\right]_1 \times \exp\left[s\left(y_{i-1(1,2)}\right)\right]_2\big)$
> > > Expression 2:
> > > $[\text{log det of jacobian}]_{2\times 2\times 1} = \begin{bmatrix}log(1)& log(1) \\log\big(\exp\left[s\left(y_{i-1(1,2)}\right)\right]_1\big) & log\big(\exp\left[s\left(y_{i-1(1,2)}\right)\right]_2\big)\end{bmatrix}$
> > >
> > > We adopt the second expression so that each term corresponds to the contribution to the log-likelihood by the dimension corresponding to that position, and sum the log-det of jacobian for all flow layers till $i^{th}$ layer, which you have correctly pointed out.
> > >
> > > Thank you for raising this confusion, we realized that expressing the above in the form of eq.9 in our manuscript might not be optimal for understanding. We will describe eq.9 in a more lucid way when the editing window is open.
> > >
> > > [RealNVP]: Density Estimation using RealNVP (https://arxiv.org/pdf/1605.08803.pdf )

---

### Author Response · Authors · 2019-11-13
**Additional experiments as per common comments from the reviewers and updated manuscript**

A common comment we received from all the reviewers was the applicability of likelihood contribution based multi-scale architecture (LCMA) for recent work on large-scale flow-based generative models, such as Glow (Kingma and Dhariwal, 2018) and flow models with ODE based density estimators such as i-ResNet (Behrmann et al., 2018), FFJORD (Grathwohl et al., 2018) and residual flows (Chen et al., 2019).

We performed additional experiments by implementing LCMA for Glow and i-ResNet and comparing it with the baseline models with CIFAR-10 dataset. We picked i-ResNet as a representative flow for the family of flow models with ODE based density estimators which employ variants of log-det computation using a power series sum of traces and hutchinson's trace estimator to estimate the individual terms in the power series. The implementation details for Glow has been included in Section 3.1 and for i-ResNet in Section 3.2 in the updated manuscript. The density estimation results for CIFAR-10 are presented in Table 2. The results (in bits/dim) are:
______________________________________________________________________________________________________________
|                      Type of multi-scale architecture                                     |   RealNVP | Glow  | i-ResNet |
______________________________________________________________________________________________________________
|              Conventional Multi-scale Architecture                                  |      3.49      | 3.35    |  3.45       |
______________________________________________________________________________________________________________
| Likelihood Contribution based Multi-scale Architecture (LCMA)   |      3.43      |  3.31  |  3.40*     |
______________________________________________________________________________________________________________
$^*$Model for i-ResNet has not fully converged

All the flow models RealNVP, Glow and i-ResNet with LCMA have better density estimation scores as compared to the baseline model which involves a static data factorization method in the multi-scale architecture. The applicability of LCMA for a wide range of flow models signifies the versatility of its application for generative flow models.

Glow: D. P Kingma and P. Dhariwal. Glow: Generative flow with invertible 1x1 convolutions, In Advances in Neural Information Processing Systems, pp. 10215–10224, 2018.
i-ResNet: J. Behrmann, W. Grathwohl, R.TQ Chen, D. Duvenaud, and J. Jacobsen, Invertible residual networks.arXiv preprint arXiv:1811.00995, 2018
FFJORD: W. Grathwohl, R. TQ Chen, J. Betterncourt, I. Sutskever, and D. Duvenaud, Ffjord:Free-form  continuous  dynamics  for  scalable  reversible  generative  models.arXiv  preprintarXiv:1810.01367, 2018
Residual Flows: R. TQ Chen, J. Behrmann, D. Duvenaud, and J. Jacobsen, Residual flows for invertible generative modeling.arXiv preprint arXiv:1906.02735, 2019

---

### Decision · Program_Chairs · 2019-12-19

**Decision:**

Reject

**Comment:**

The authors propose a multi-scale architecture for generative flows that can learn which dimensions to pass through more flow layers based on a heuristic that judges the contribution to the likelihood. The authors compare the technique to some other flow based approaches. The reviewers asked for more experiments, which the authors delivered. However, the reviewers noted that a comparison to the SOTA for CIFAR in this setting was missing. Several reviewers raised their scores, but none were willing to argue for acceptance.